# Circulating whole genome miRNA expression corresponds to progressive right ventricle enlargement and systolic dysfunction in adults with tetralogy of Fallot

**Chad S. Weldy**[1], **Saad Ali Syed**[2], **Myriam Amsallem**[1,3], **Dong-Qing Hu**[3], **Xuhuai Ji**[4], **Rajesh Punn**[3], **Anne Taylor**[3], **Brittany Navarre**[3], **Sushma Reddy**[3]*

**1** Division of Cardiology, Department of Medicine, Stanford University, Stanford, California, United States of America, **2** Stanford University School of Medicine, Stanford, California, United States of America, **3** Division of Cardiology, Department of Pediatrics, Stanford University, Stanford, California, United States of America, **4** Human Immune Monitoring Center and Functional Genomics Facility, Stanford University, Stanford, California, United States of America

* sureddy@stanford.edu

## Abstract

### Introduction

The adult congenital heart disease population with repaired tetralogy of Fallot (TOF) is subject to chronic volume and pressure loading leading to a 40% probability of right ventricular (RV) failure by the 3rd decade of life. We sought to identify a non-invasive signature of adverse RV remodeling using peripheral blood microRNA (miRNA) profiling to better understand the mechanisms of RV failure.

### Methods

Demographic, clinical data, and blood samples were collected from adults with repaired TOF (N = 20). RNA was isolated from the buffy coat of peripheral blood and whole genome miRNA expression was profiled using Agilent's global miRNA microarray platform. Fold change, pathway analysis, and unbiased hierarchical clustering of miRNA expression was performed and correlated to RV size and function assessed by echocardiography performed at or near the time of blood collection.

### Results

MiRNA expression was profiled in the following groups: 1. normal RV size (N = 4), 2. mild/moderate RV enlargement (N = 11) and 3. severe RV enlargement (N = 5). 267 miRNAs were downregulated, and 66 were upregulated across the three groups (fold change >2.0, FDR corrected p<0.05) as RV enlargement increased and systolic function decreased. qPCR validation of a subset of these miRNAs identified increasing expression of miRNA 28-3p, 433-3p, and 371b-3p to be associated with increasing RV size and decreasing RV systolic function. Unbiased hierarchical clustering of all patients based on miRNA expression demonstrates three distinct patient clusters that largely coincide with progressive RV

**Data Availability Statement:** All relevant data are within the manuscript and its Supporting Information files.

**Funding:** SR was supported by grants from the NIH K08 HL127277, U.S. Department of Defens PR151448, Reddy Foundation, and American Heart Association 16GRNT31200008. MA was supported by a Stanford Cardiovascular Institute – MHRCI seed grant. The funders had no role in study design, data collection and analysis, decision to publish, or preparation of the manuscript.

**Competing interests:** The authors have declared that no competing interests exist.

enlargement. Pathway analysis of dysregulated miRNAs demonstrates up and downregulation of cell cycle pathways, extracellular matrix proteins and fatty acid synthesis. HIF 1α signaling was downregulated while p53 signaling was predicted to be upregulated.

## Conclusion

Adults with TOF have a distinct miRNA profile with progressive RV enlargement and dysfunction implicating cell cycle dysregulation and upregulation in extracellular matrix and fatty acid metabolism. These data suggest peripheral blood miRNA can provide insight into the mechanisms of RV failure and can potentially be used for monitoring disease progression and to develop RV specific therapeutics to prevent RV failure in TOF.

## Introduction

Congenital heart disease (CHD) is the most common form of birth defects, with an estimated prevalence of 11.9 per 1000 children [1]. Due to tremendous growth in surgical and catheter-based interventions for treating patients with CHD, infant and childhood mortality has markedly declined [2]. As a result, there are now more adults than children living with CHD [3]. Heart failure is the most common cause of death in the adult CHD (ACHD) population [4]. The incidence of heart failure admissions in the ACHD population is estimated at 1.2 per 1000 patient-years, a greater than 10-fold higher incidence rate for a comparative aged population without CHD [5]. The risk of heart failure in ACHD population increases with increasing complexity of the heart defect such as in single ventricle, tetralogy of Fallot (TOF), and transposition of the great arteries after atrial switch operation [6, 7]. TOF is the most common form of cyanotic CHD, seen in 3.5% of infants born with CHD and is the largest growing group of complex ACHD patients [8]. Despite successful surgical repair, many patients have chronic volume and pressure loading leading to a 40% probability of right ventricular (RV) failure by the 3rd decade of life [9]. In addition, standard medical therapies used to treat left heart failure, such as ACEI/ARB and β-blocker therapy, have not been shown to provide benefit [10–13].

Replacement of the pulmonary valve to correct chronic RV volume or pressure overload due to pulmonary regurgitation or RV outflow tract obstruction respectively can reduce RV size. However, the risk of ventricular tachycardia, sudden death, exercise capacity, functional heart failure class, long term RV function, and quality of life remain major clinical concerns [14]. Furthermore, the optimal timing of pulmonary valve replacement to prevent adverse RV remodeling and thereby improve long term outcomes is unknown [15]. Therefore, there is a need to better understand the cellular mechanisms which lead to progressive RV enlargement and dysfunction in TOF patients.

MicroRNA (miRNA) are noncoding RNA ~22 nucleotides in length which regulate gene expression through base-pairing with mRNA leading to mRNA degradation or translational inhibition. There is growing interest in circulating miRNA as biomarkers of heart failure [16, 17]. We have previously reported that miRNA expression is dynamically regulated in experimental RV failure [18]. Circulating miRNA profiles have been found to be distinct in patients with TOF, ventricular septal defect, and transposition of the great arteries after atrial switch operation when compared to healthy controls [19–22], but there has been limited to no association between miRNA expression and clinically significant endpoints in these patients.

We hypothesized that miRNA profiles in circulating cells would provide insight into the mechanism of disease progression which could then aid in risk stratification and early

prediction of RV failure in patients with TOF. We show that global peripheral blood miRNA isolated from the buffy coat are significantly altered in TOF patients with RV volume overload and RV enlargement and correlate with RV size and function.

## Methods

### Patient population

This study was approved by the Institutional Review Board (IRB) of Stanford University Medical Center. Inclusion criteria included age > 18 years, diagnosis of TOF with or without pulmonary atresia or major aortopulmonary collateral arteries (MAPCA) s/p complete surgical repair. Exclusion criteria included extracardiac co-morbidities including hepatic, renal or pulmonary disease, and intercurrent illness. Twenty adult patients with repaired tetralogy of Fallot were recruited. Patient characteristics are shown in Table 1. Patients were recruited in the outpatient clinic at the time of a routine, annual follow up visit which included a transthoracic echocardiogram (TTE). Patients were not included if a clinic visit and TTE was urgently indicated due to rapidly progressive heart failure, arrhythmias, or concern for acute conduit/valvular abnormalities due to endocarditis. To evaluate distinct miRNA expression changes, patients were grouped into 3 categories based on the qualitative assessment of RV size on transthoracic echocardiogram (TTE) by the reading cardiologist and subsequently confirmed by quantitative measurement of RV size by an independent cardiologist blinded to the patient and miRNA data: (1) "normal" RV size, (2) "mild/moderate" RV enlargement, and (3) "severe" RV enlargement.

### Echocardiography

Digitized studies were acquired using Philips ultrasound systems. All measures were averaged over 3 cycles and analyzed according to the latest guidelines by a blinded certified level 3

**Table 1. Characteristics of adult patients with repaired tetralogy of Fallot included in study.**

| | All | Normal | Mild/Moderate | Severe |
|---|---|---|---|---|
| # of patients | 20 | 4 | 11 | 5 |
| Age at blood draw (Median, IQR) | 36.9 (26.5–43.8) | 29.0 (22.4–36.6) | 35.8 (26.3–40.7) | 46.5 (42.7–51.2) |
| # female | 9/20 | 0/4 | 6/11 | 3/5 |
| BMI (Median, IQR) | 25.6 (22.8–29.6) | 26.5 (25.6–28.1) | 24.7 (22.5–28.9) | 23.5 (22.7–31.4) |
| NHYA (Median, range) | 1.8 (1–2) | 2 | 2 | 2 |
| Number of surgeries (Median, range) | 2 (1–4) | 2 (1–3) | 2 (1–4) | 2 (1–3) |
| Age at complete repair (Median, IQR) | 5 (1.75–9.25) | 6 (1.75–18) | 5 (2–9.5) | 5 (1–6) |
| # prior palliative shunt | 7/20 | 1/4 | 3/11 | 3/5 |
| # transannular patch | 16/20 | 3/4 | 8/11 | 5/5 |
| # RV to PA conduit | 6/20 | 1/4 | 4/11 | 1/5 |
| # pulmonary valve replacement | 3/20 | 1/4 | 1/11 | 1/5 |
| # on diuretic | 3/20 | 1/4 | 1/11 | 1/5 |
| # on beta blocker | 7/20 | 1/4 | 3/11 | 3/5 |
| Na (mmol/L) (Median, IQR) | 139 (138–140) | 138 (137.5–141) | 139 (137.5–140.5) | 140 (139–141) |
| Cr (mg/dL) (Median, IQR) | 0.8 (0.71–0.85) | 0.75 (0.67–0.8) | 0.79 (0.7–0.85) | 0.86 (0.8–0.9) |
| AST (U/L) (Median, IQR) | 24 (20–32) | 24 (22–45) | 23 (15–32) | 30 (22–32) |
| ALT (U/L) (Median, IQR) | 33 (25–77) | 46 (40–72) | 26 (24–77) | 30 (29–52) |
| Alk Phos (U/L) (Median, IQR) | 76 (62–88) | 104 (88–110) | 71 (54–86) | 78 (62–79) |
| Total Bilirubin (mg/dL) (Median, IQR) | 0.6 (0.4–0.8) | 0.4 (0.3–0.5) | 0.6 (0.5–0.8) | 0.4 (0.4–1) |

reader (M.A.). Indices of right heart remodeling and function were measured on the focused RV apical 4-chamber view; measures included end-diastolic (RVEDA) and end-systolic areas (RVESA), maximal right atrial area (RA), RV fractional area change, and RV free wall Lagrangian longitudinal strain (RVLS). RVLS was measured from mid-endocardial end-diastolic and end-systolic manually traced lengths (as the software-based speckle-tracking method was not possible in all patients due to the quality of the images) and calculated as (end-systolic length−end-diastolic length)/end-diastolic length as previously validated [23]. Frame rate for RVLS acquisition ranged from 39–70 frames/second. RAP was estimated from the inferior vena cava size and collapse according to the recent 2015 American Society of Echocardiography guidelines. RV systolic pressure from the tricuspid regurgitation maximal velocity and estimated RAP as previously published [24]. The severity of the tricuspid regurgitation was qualitatively classified into 4 grades. Left ventricular ejection fraction was measured from the apical 4-chamber view by method of disk.

Cardiac MRI (cMRI) imaging was not available for all patients and therefore comparisons of RV size and function to miRNA signature based on cMRI were not performed. Of 20 patients recruited, 12 patients had a cMRI but there were only 4 patients where cMRI was done at or near the time of echocardiogram and blood draw (S1 Table in S1 File). Therefore, assessment of RV size and function based on TTE imaging done at the time of blood sample collection was used. RV size and function by TTE was compared with the gold standard cMRI derived RV size and function in the 4 patients with recently available cMRI.

## MiRNA expression profiling

After consent was obtained, a peripheral blood sample was collected, spun down and the buffy coat was collected and stored at -80˚C for future miRNA isolation. Total RNA was isolated from frozen buffy coats (200 ul) using TRIzol (Invitrogen, Carlsbad, CA, USA) and purified using RNeasy Mini Kit (QIAGEN, Dusseldorf, Germany) according to the manufacturer's instructions. RNA quality and quantity were measured using a QIAxpert (QIAGEN, Cat. #9002340, Germany). RNA integrity was determined by the Bioanalyzer 2100 NANO analysis (Agilent, Santa Clara, USA). MiRNA labeling and array hybridization were carried out according to Agilent's protocol for the miRNA Microarray System with miRNA Complete Labeling and Hyb Kit (Agilent, p/n 5190–0456). Briefly, 100 ng of total RNA was used for miRNA labeling with cyanine 3 then hybridized onto the SurePrint custom G3 miRNA Microarray (8x60k, p/n G4871A) then washed and scanned according to the manufacturer instructions. Data were extracted using Agilent Feature Extraction (FE) Software. Microarray normalization was performed by GeneSpring GX 14.9.1 software. All arrays were normalized by using the 75th percentile method [25].

## miRNA microarray and pathway analysis

MiRNA expression analyses were performed using GeneSpring GX 14.9.1 software (Agilent). Normalized data between the 20th and 100th percentile with detected probes were used for further analysis. Quality control was performed, after which ANOVA with Benjamini Hochberg multiple testing correction was applied. Significantly altered miRNA was determined based on corrected P value < 0.05 and fold change between any group >2.0. MiRNA pathway analysis and target prediction was performed using Diana Tools miRPath v.2.0 [26] with TarBase V7.0. KEGG pathways were deemed significant with FDR correction and P<0.05.

## Biomark microfluidic qPCR array

Gene expression of 5 candidate miRNA identified by microarray, miR 28-3p, miR 34b-3p, miR 371b-3p, miR 433-3p, and miR 451b were validated by qPCR using Fluidigm Biomark

multiplexing qPCR using target-specific stem-loop reverse transcription (RT) primers for 3'
templates. Fold change was evaluated using deltaCT method from "normal" RV group. 10 ng
of total RNA was reverse transcribed to cDNA at 50˚C for 15 minutes using the High Capacity
Reverse Transcription kit (ABI). RT was performed directly on a 96-well PCR plate (ABI) con-
taining lysis buffer (Invitrogen) by using SuperScript III One-Step RT-PCR System with Plati-
numTaq (CellDirect kit, Invitrogen). PreAmp was performed on a thermocycler using the
TaqMan PreAmp Master Mix Kit (Invitrogen) added to cDNA and pooled Taqman assays. RT
enzyme was inactivated and the Taq polymerase reaction was started by bringing the sample
to 95˚C for 2 minutes. The cDNA was preamplified by denaturing for 10 (total RNA) to 18
(single cell) cycles at 95˚C for 15 seconds, annealing at 60˚C for 4 minutes. The resulting
cDNA product was diluted 1:2 with 1x TE buffer (Invitrogen). 2X Applied Biosystems Taqman
Master Mix, Fluidigm Sample Loading Reagent, and preamplified cDNA were mixed and
loaded into the 48.48 Dynamic Array (Fluidigm) sample inlets, followed by loading 10X assays
into the assay inlets. Manufacturer's instructions for chip priming, pipetting, mixing, and load-
ing onto the BioMark system were followed. Real-time PCR was carried out with the following
conditions: 10 min at 95˚C, followed by 50 cycles of 15 sec at 95˚C and 1 min at 60˚C. Data
were analyzed using Fluidigm software. All reactions were performed in triplicate.

## Statistical analysis

Echocardiographic and qPCR data of continuous variables are presented as mean with stan-
dard error and geometric mean with geometric standard error respectively. Differences
between normal RV, mild to moderate RV enlargement and severe RV enlargement were ana-
lyzed using one-way ANOVA with Tukey's multiple-comparison testing. Linear regression
between two continuous variables was compared using simple linear regression modeling.
$p < 0.05$ was considered significant using Prism v8.3.1 (GraphPad Software LLC). The funders
had no role in study design, data collection and analysis, decision to publish, or preparation of
the manuscript.

# Results

## Patient characteristics and echocardiographic analysis

Patient demographics are shown in Table 1. The median patient age was 36.9 years at the time
of blood draw (interquartile range (IQR) 26.5–43.8 years). Nine of 20 patients (45%) were
female. Median BMI was 25.6 (IQR 22.8–29.6). All patients had either NYHA class I or II
symptomology by chart review. All patients were diagnosed with TOF, two of whom had con-
current major aortopulmonary collateral arteries (MAPCAs). Patients had undergone a
median of 2 (range 1–4) cardiac surgeries prior to recruitment (Table 1). Median age of com-
plete repair was 5 years of age (range 1–42, IQR 1.75–9.25). Most common sequence of repair
for these patients included an initial Blalock Taussig (BT) Shunt as a neonate with subsequent
complete repair with VSD closure and transannular patch in childhood, although 3 of 20
patients had late repairs (no surgical management until age > 10). Three of 20 patients were
on diuretic therapy and 7 of 20 patients were on β-blocker therapy for arrhythmias, which
were well controlled at the time of recruitment. No patients were on ACE inhibitors. Data
were collected from transthoracic echocardiograms performed at or near the time of blood col-
lection. Of 20 patients, 4 patients were identified as having normal RV size, 11 patients were
identified as having mild/moderate RV enlargement, and 5 were identified as having severe
RV enlargement. Patients in the "normal" RV size group had more females and were younger.
Of the 3 patients with late repair, 1 was in the "normal" RV size group and 2 were in the "mild/
mod" RV size group. RV end-diastolic area (RVEDA), RV end-systolic area (RVESA), and

right atrial (RA) area increased from normal RV size to mild-moderate RV enlargement to severe RV enlargement (Fig 1A–1C). There was a trend towards a decrease in RV fractional area change (RVFAC) and RV longitudinal strain (RVLS) between normal and severe groups but this did not reach significance (Fig 1D and 1E). No patient had grade 4 tricuspid regurgitation. There was no significant difference in LV ejection fraction (LVEF) between groups (Fig 1F). The RVEDA (cm2), RVESA (cm2) and RVFAC (%) by echocardiogram were compared to the MRI RVEDV (ml), RVESV (ml) and RVEF (%) respectively in 4 patients where cMRI

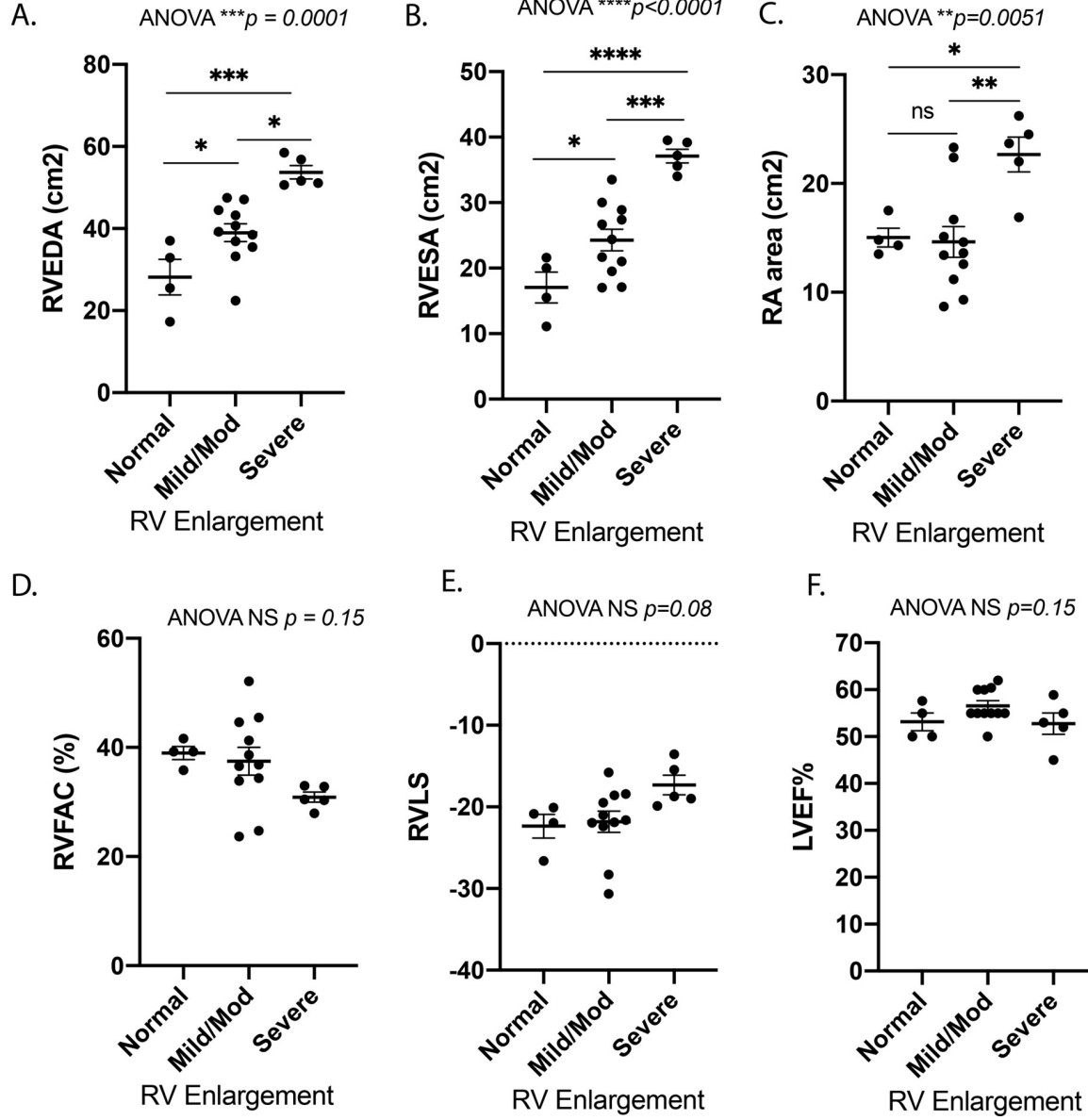

**Fig 1. Echocardiographic characteristics by patient group based on initial qualitative RV assessment of "normal", "mild/mod" and "severe" RV enlargement.** (A) RV end diastolic area (cm2) (RVEDA) and (B) RV end systolic area (cm2) (RVESA) increased with mild-moderate and severe RV enlargement, (C) right atrial area (cm2) increased with severe RV enlargement, (D) RV fractional area change (RVFAC) showed a trend toward a decrease with severe RV enlargement, (E) RV free wall Lagrangian longitudinal strain (RVLS) showed a trend toward an increase with severe RV enlargement, and (F) LV ejection fraction (LVEF%) was similar in all groups, N = 4-11/group. RV–right ventricle, LV–left ventricle. Data is shown as mean±SEM. Comparison with ANOVA and Tukey's multiple comparisons test.

was available. There is a strong and significant correlation between the echocardiogram and the cMRI assessment of RV size (measures of RVEDA or RVESA vs RVEDV or RVESV respectively: R = 0.83, $p<0.01$) and a trend of association for measure of RV systolic function (RVFAC vs RVEF: R = 0.77, $p = 0.22$) (S1 Table in S1 File).

## Global miRNA changes correspond to progressive RV enlargement in adult patients with TOF

Of 20 patients, 14 had blood samples collected within 24 hours of a clinically indicated transthoracic echocardiogram. Of the remaining 6 patients, 3 had blood samples collected within 7 days of a clinically indicated transthoracic echocardiogram, 1 within 14 days, and 2 within 5 months. Of the two patients with greater duration between blood sample and echocardiogram, RV size and function were stable based on follow-up echocardiogram. Global miRNA microarray profile decreased as RV size increased (Fig 2A). 333 miRNA were significantly dysregulated between patients with normal RV size, mild-moderate RV enlargement, and severe RV enlargement (fold

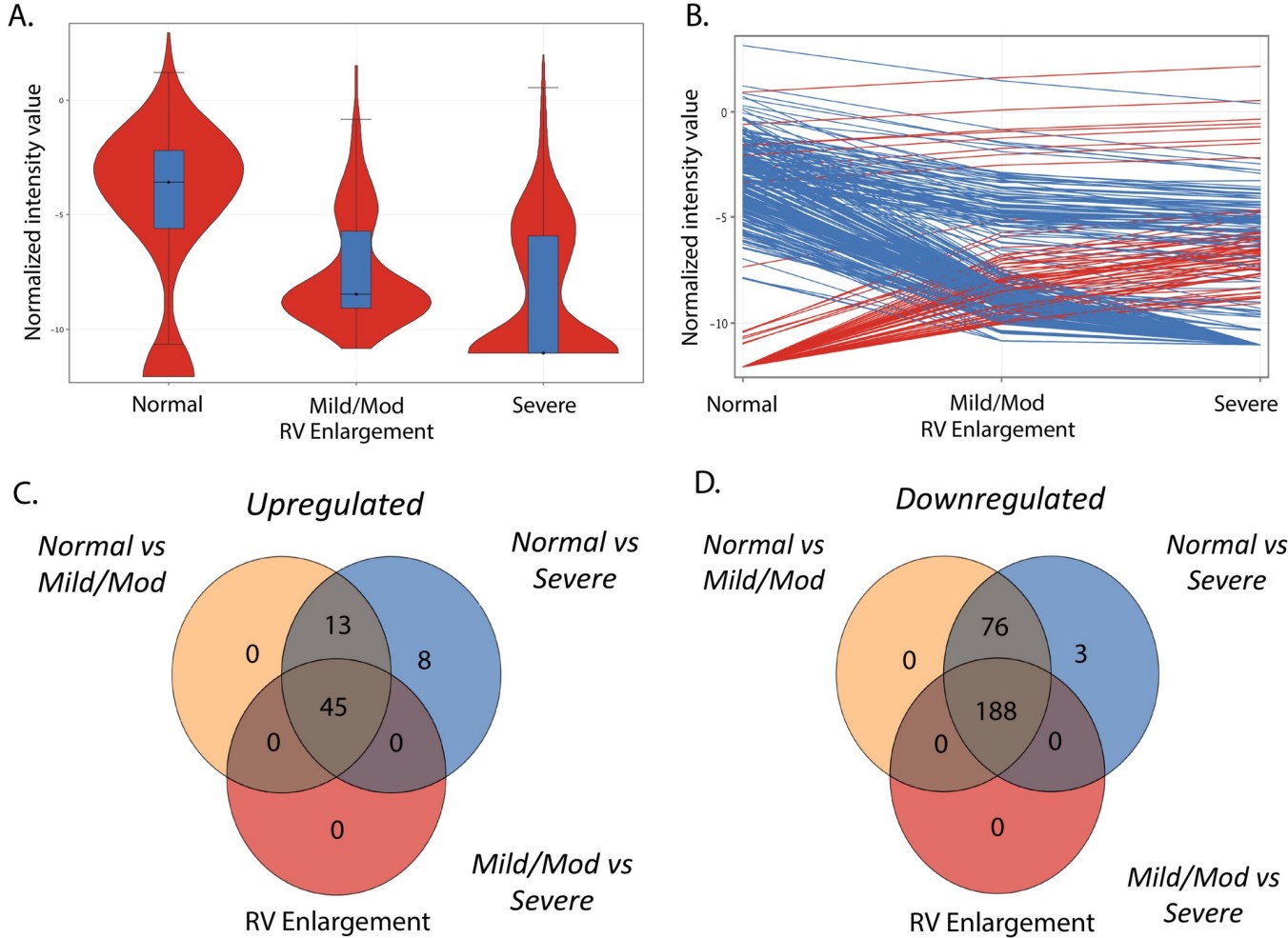

**Fig 2. Global miRNA expression in tetralogy of Fallot changes with increasing RV size.** (A). Violin plot demonstrates decreasing miRNA expression with increasing RV enlargement with greater variability in miRNA expression in severe RV enlargement, N = 4-11/group. (B) Individual significantly dysregulated miRNA (FDR <0.05) change with increasing RV enlargement. (C) (D) Venn diagram analyses of upregulated and downregulated miRNAs between groups show that the same miRNAs are dysregulated across groups. RV–right ventricle.

change > 2.0, FDR corrected p< 0.05), 267 miRNA were downregulated and 66 were upregulated (Fig 2B) (S2 and S3 Tables in S1 File). Interestingly, these miRNAs progressively changed with RV size (Fig 2C and 2D). Intriguingly, there was remarkable concordance where increasing miRNA between the normal to mild/moderate RV enlargement groups appeared to be largely the same miRNA increased between the normal to severe and mild/moderate to severe RV enlargement groups. A similar finding was seen for decreasing miRNA (Fig 2C and 2D).

## Unbiased hierarchical clustering and principal component analysis

We next evaluated whether unbiased hierarchical clustering of patients based on global miRNA expression would correspond to RV size and or function. We first performed unbiased clustering between the 3 predetermined groups (Normal RV size, mild/moderate RV enlargement, and severe RV enlargement), where we observed that the normal group clustered away from the mild/mod and severe groups (Fig 3A). We then performed hierarchical clustering among all 20 patients, where patients clustered into 3 distinct groups (Fig 3B): group 1 was predominantly "normal" and "mild/moderate", group 2 was predominantly "mild/moderate" and some "severe", and group 3 appeared to be equally "mild/moderate" and "severe". Principal component analysis of all 20 patients revealed distinct separation between patients, where patients with "normal" RV size appeared to cluster together, and those with "severe" RV enlargement

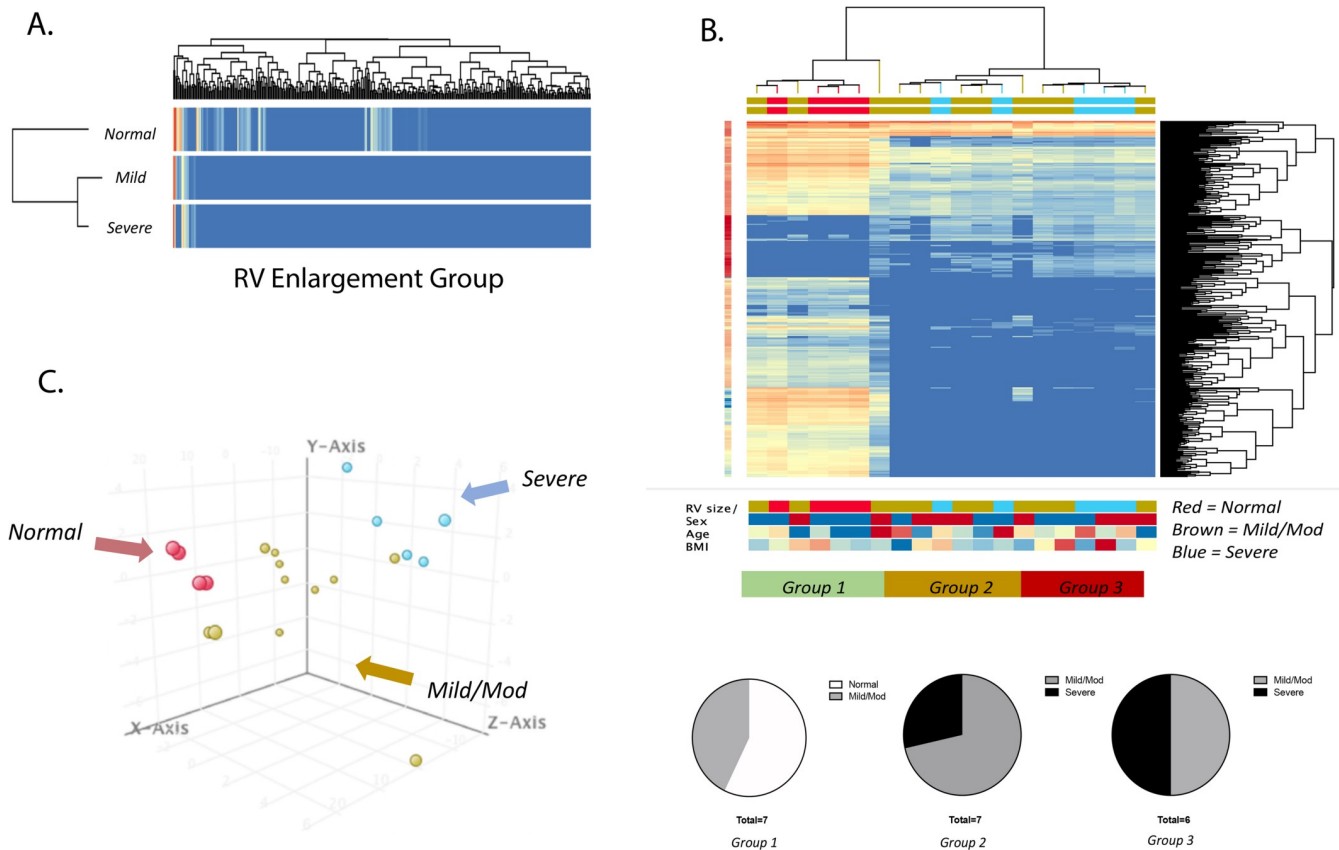

**Fig 3. Hierarchical clustering of patients based on miRNA expression shows relationship to RV size.** (A) Heatmap of groups based on echocardiography into normal RV size (N = 4), mild-moderate (N = 11) and severe RV enlargement (N = 5). (B) Unbiased hierarchical clustering based on differentially expressed miRNA identifies three distinct groups: Group 1—patients with mostly normal RV size, Group 2 –patients with mostly mild-moderate RV enlargement and Group 3 –patients with mild-moderate and severe RV enlargement (ANOVA FDR < 0.05). (C) Principal component analysis of individual patients demonstrates clustering of patients based on miRNA global profile (red = normal, brown = mild/mod, blue = severe). RV–right ventricle.

appeared to cluster together, and those with "mild/moderate" RV enlargement appeared to reside between or to cluster closer to "normal" or "severe" enlargement groups (Fig 3C). We observed a trend toward increasing RVEDA and an increase in RVESA ($p$ = 0.04) between group 1 to 3 (S1A and S1B Fig in S1 File), showing unbiased clustering grouped patients by RV size. We also observed a trend toward a larger right atrial area, lower RVFAC and RVLS (S1C–S1E Fig in S1 File). LVEF was unchanged between the groups (S1F Fig in S1 File).

## MiRNA dysregulated with RV enlargement predict derangements in cell cycle, extracellular matrix, metabolism, and cell death

To better understand the implication for the observed global miRNA expression profile changes, we performed pathway analysis on both upregulated and downregulated miRNA using Diana Tools software. We observed 42 pathways to be significantly dysregulated by upregulated miRNAs and 58 pathways to be significantly dysregulated by downregulated miRNAs (S4 and S5 Tables in S1 File). Clustering dendrogram analysis of dysregulated pathways for upregulated miRNAs show that pathways appear to cluster into 4 main groups suggesting downregulation in, 1) pathways in cancer, proteoglycans in cancer, and viral carcinogenesis, 2) adherens junction, HIF-1 signaling, and glioma, 3) cell cycle, hippo signaling, and estrogen signaling pathway, and 4) extracellular matrix (ECM)-receptor interaction, fatty acid biosynthesis, and fatty acid metabolism (Fig 4A). Dysregulated pathways for downregulated miRNAs also appear to cluster into 4 main groups suggesting upregulation in, 1) cell cycle, p53 signaling, and RNA transport, 2) TGF-β signaling pathway, lysine degradation, and hippo signaling, 3) adherens junction, FoxO signaling pathway, and protein processing in endoplasmic reticulum, 4) ECM-receptor interaction, fatty acid biosynthesis, fatty acid metabolism (Fig 4B).

## Cell cycle pathways are the most dysregulated in RV enlargement in TOF

Cell cycle pathways appear to be the most significantly dysregulated in both the up and downregulated pathways. Among the upregulated miRNA, 13 miRNA were involved in cell cycle pathways, including miRNAs 28-3p, 149-3p, 196a-3p, 431-3p, and 675-5p, predicting downregulation of 51 target genes including SMAD3 and 4, TGF-β1 and 2, E2F family transcription factors including E2F1, E2F2, and E2F5, and cell division cycle family of regulatory proteins including CDC6, 20, 23, and 27. For downregulated miRNA, 34 miRNA were identified as being involved in cell cycle pathway, including miRNAs 19b-1-5p, 20b-3p, 122-5p, 130a-5p, and 197-3p, predicting upregulation of 83 target genes similar to that seen in upregulated miRNA. Interestingly, these downregulated miRNAs are also known to increase the expression of target genes HDAC1, HDAC2, and RB1 which increase global chromatin accessibility, increase cardiac fibrosis and decrease cardiac function [27, 28]. The overall weight of dysregulated miRNA in cell cycle pathways suggests a net upregulation of this pathway. The TGF-β signaling pathway was also found to be significantly dysregulated by both the up and downregulated miRNA, driven by mRNA targets including TGF-β, SMAD and E2F families, and also BMPR1B, BMPR2, MYC, and ROCK1. Therefore, the predicted mRNA targets suggest upregulation of many components of the cell cycle pathway, mostly the extracellular matrix which may be countered by downregulation of many components of the same pathway.

## HIF-1α pathway is downregulated in RV enlargement in TOF

Upregulated miRNA seen in RV enlargement predict downregulation of HIF1 signaling ($p$ = 7.7E-4) which may suggest an inhibition of this pathway. There were 11 miRNA found to be involved in this pathway, including miRNAs 28-3p, 138-5p, 143-5p, 431-3p, and 671-5p. There were 36 significant mRNA targets identified, which include HIF1A, NFKB1, VEGFA,

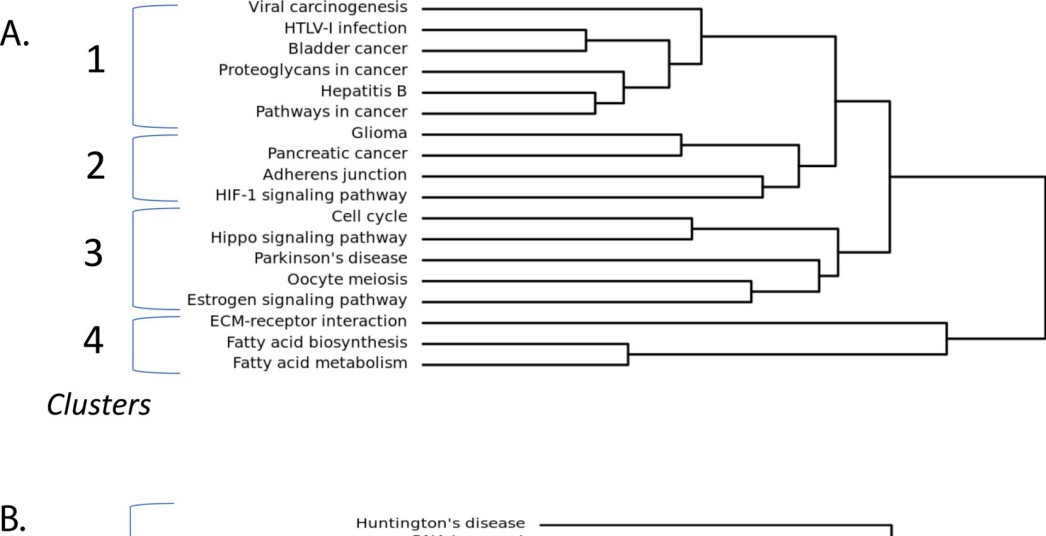

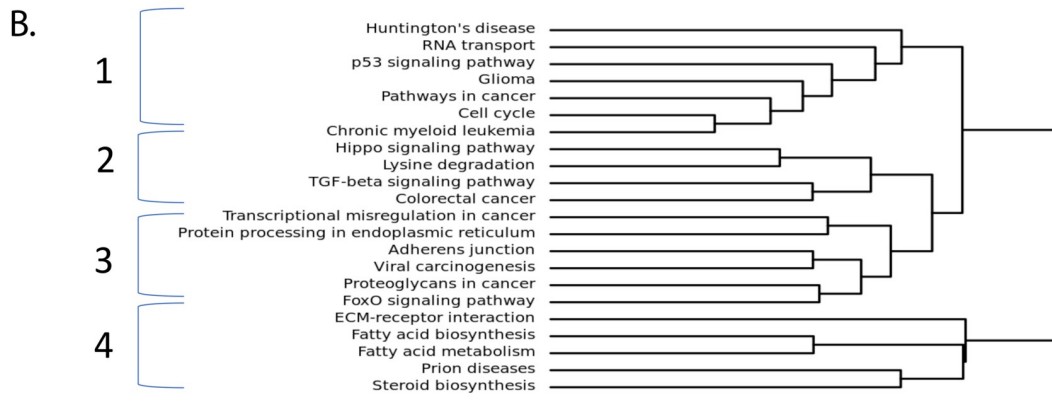

**Fig 4. Pathway analysis shows distinct biological pathway clusters.** Clustering of significantly dysregulated pathways for (A) upregulated miRNA and (B) downregulated miRNA. Pathways separate into 4 main clusters, both up and down regulated pathways include components of cell cycle both and fatty acid metabolism. Targets of upregulated miRNA include HIF1 signaling pathway. Targets of downregulated miRNA include TGF-β signaling pathway. FDR < 0.05.

IL6R, TIMP1, EGFR, PRKCA, and BCL2. This finding may suggest an inhibition of these mRNA targets involved in angiogenesis in response to hypoxia.

### Fatty acid metabolism is dysregulated in RV enlargement in TOF

Pathway analysis also revealed fatty acid biosynthesis to be significantly dysregulated by both up (p = 4.79E-09) and downregulated miRNA (4.76E-06). For upregulated miRNA, this pathway is driven by miRNA 604 which targets fatty acid synthase. There were 9 downregulated miRNA found to be involved in this pathway, including miRNAs 122-5p, 149-5p, 185-3p, 197-3p, and 629-3p which target fatty acid synthase, acetyl-coA carboxylase A and B, malonyl-CoA-acyl carrier protein transacylase, and long-chain acyl-CoA synthetase isoform 1 and 4. This may suggest a net upregulation in fatty acid metabolism to maintain the metabolic demands of the heart (S2 Fig in S1 File).

### qPCR validation of miRNA microarray data

qPCR validation of miRNA microarray results was performed on 5 select miRNA which have been previously described in heart failure, cell cycle progression, coronary artery disease, or

vascular function [18, 29–34]: miRNA 28-3p, miRNA 433-3p, miRNA 371b-3p, miRNA 34b-3p and miRNA 451b. MiRNAs 28-3p, 371b-3p, and 433-3p expression increased in patients with mild-moderate RV enlargement and severe RV enlargement compared to those with normal RV size (S3A, S3C and S3D Fig in S1 File), miRNA 451b expression decreased similar to the microarray results (S3E Fig in S1 File) while miRNA 34b-3p expression did not change (S3B Fig in S1 File).

## miRNA expression correlates with of RV size and function

We further evaluated the relationship between the miRNAs validated by qPCR and echocardiographic RV size and function using linear regression analysis. There was a linear relationship between increasing miR 28-3p expression and increasing RVEDA ($R^2$ = 0.23, $p$ = 0.03) and RVESA (Fig 5A and 5B) ($R^2$ = 0.32, $p$ = 0.009). Increasing miRNA 28-3p expression was also associated with decreasing RVFAC and RVLS ($R^2$ = 0.17, $p$ = 0.06 and $R^2$ = 0.22, $p$ = 0.036, respectively) (Fig 5C and 5D). Increasing miRNA 433-3p expression was associated with a trend of significance with increasing RVEDA and a significant relationship to RVESA (Fig 5E and 5F) ($R^2$ = 0.17, $p$ = 0.06 and $R^2$ = 0.24, $p$ = 0.028, respectively). There are similar trends of a relationship of increasing miRNA 433-3p and decreasing RVFAC and RVLS ($R^2$ = 0.12, $p$ = 0.13 and $R^2$ = 0.16, $p$ = 0.07, respectively) (Fig 5G and 5H). Increasing miRNA 371b-3p expression was not associated with increasing RV size but was associated with decreasing RVFAC ($R^2$ = 0.19, $p$ = 0.049) (S4A–4D Fig in S1 File). MiRNA 451b expression was downregulated and was weakly associated with increasing RVEDA but not with RVESA, RVFAC, or RVLS (S4E–S4H Fig in S1 File). Therefore, many of the validated miRNAs were associated with increasing RV size and decreasing RV function.

## Discussion

The adult congenital heart disease population with repaired tetralogy of Fallot (TOF) is rapidly growing in numbers but are subject to a high incidence of RV failure due to chronic volume and pressure loading [2, 6, 7, 9, 35]. Replacement of the pulmonary valve improves pulmonary regurgitation or RV outflow tract obstruction and can reduce RV size. However, the risk of arrhythmias, sudden death and long term RV function are not improved [14] suggesting that irreversible myocardial remodeling has occurred prior to pulmonary valve replacement. The cellular processes leading to irreversible myocardial damage is unknown. We evaluated a non-invasive blood-based miRNA signature in circulating cells to better understand the mechanisms of RV failure and to risk-stratify disease severity. We have identified a unique miRNA signature as the RV enlarges and becomes dysfunctional in patients with TOF. This unique miRNA signature demonstrates both up and downregulation of various components of cell cycle pathways, extracellular matrix proteins and fatty acid synthesis, progressive upregulation in cell death pathways, and downregulation in HIF-1α signaling. These findings highlight the potential for blood miRNA to be used for investigating molecular mechanisms of RV failure in TOF as well as for discovery of novel biomarkers of RV failure.

MiRNA are critical in mediating complex transcriptional changes and govern deleterious as well as protective gene expression responses to stress. MiRNA changes in peripheral blood have been utilized to identify potential novel biomarkers of cardiovascular disease including heart failure [16, 17], congenital heart disease [20, 21], pulmonary hypertension [36], and sudden cardiac death [37]. Circulating miRNA profiles can also dynamically change with pulmonary hypertension secondary to CHD [36]. In these investigations, miRNA has been measured from either plasma or serum, where presumably stable miRNAs are secreted from cells and can reflect miRNA expression changes in disease. Alternatively, some investigators have

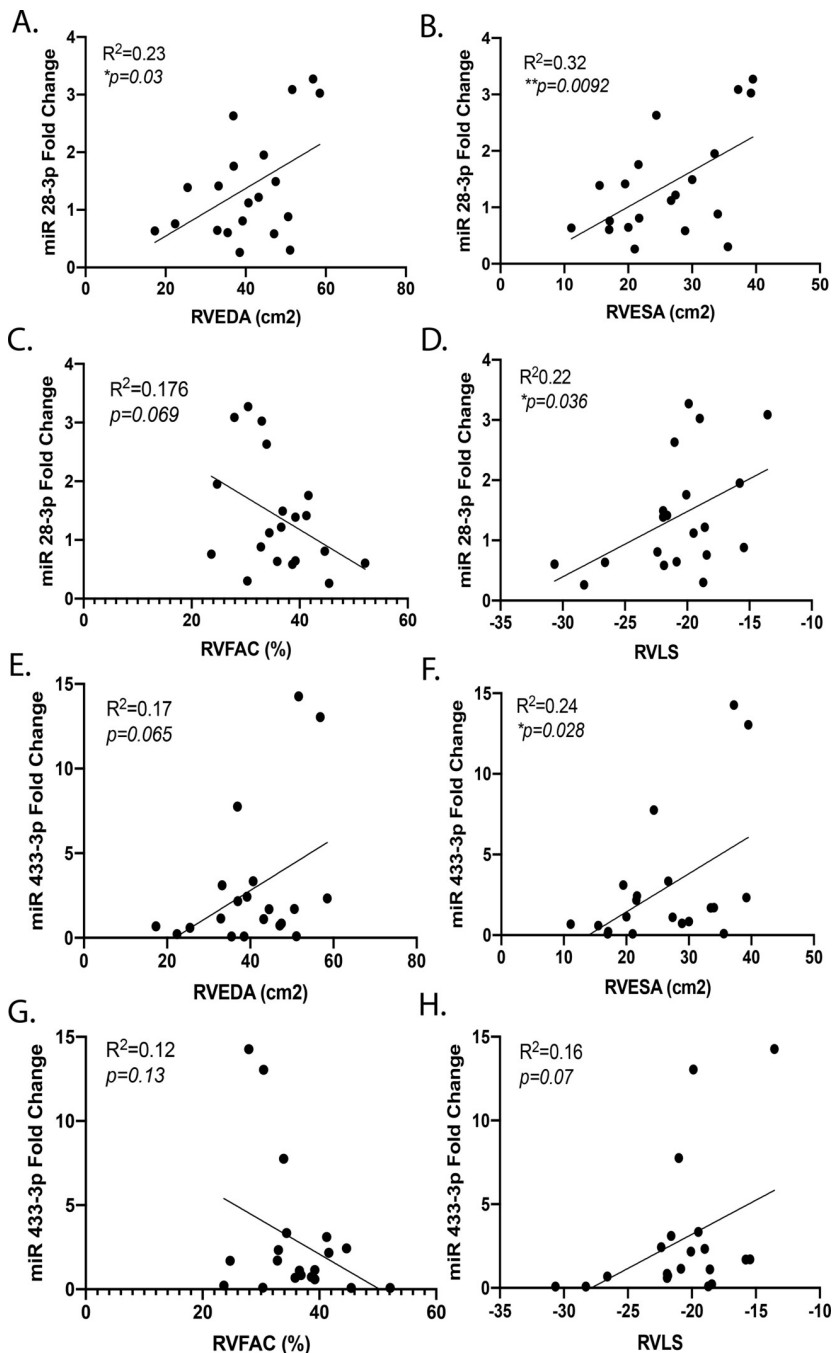

**Fig 5. miRNA 28-3p and 433-3p have linear relationships to RV size and systolic function.** Increasing expression of miR 28-3p is associated with increased (A) RVEDA and (B) RVESA, (C) a trend toward decreasing RVFAC and, (D) increasing RVLS. Increasing expression of miR 433-3p is associated with increasing (E) RVEDA (p = 0.07), (F) RVESA and (H) RVLS (p = 0.07). (G) miR 433-3p expression is not associated with RVFAC. Data is shown as dot plot with simple linear regression, *p<0.05,**p<0.01. RV–right ventricle, RVEDA–RV end diastolic area, RVESA–RV end systolic area, RVFAC–RV fractional area change, RVLS–RV longitudinal strain.

evaluated whole blood, combining miRNA isolated from plasma and intracellular miRNA from circulating cells. Importantly, these two methods have shown different results, where miRNA from whole blood appears to show a more significant and consistent change in

miRNA expression levels compared to miRNA isolated from serum [19, 21]. We therefore evaluated miRNA in circulating cells (buffy coat) in the blood and discovered heightened miRNA expression across hundreds of miRNAs compared to any other report to date. In addition, despite the small patient sample, we observed a robust, dynamic change in miRNA expression with increasing RV enlargement and systolic dysfunction. The buffy coat comprises of circulating inflammatory cells namely the white blood cells and platelets. Therefore, the heightened miRNA expression in these cells may implicate a systemic inflammatory state in repaired TOF patients even during the stage of RV enlargement prior to the development of RV failure providing insight into the mechanisms mediating RV failure.

The mechanism by which miRNA expression changes in inflammatory cells in the peripheral blood in response to progressive RV enlargement reflect and systolic dysfunction is unclear. One possibility is that as RV enlargement progresses, local and systemic stress responses are activated including adrenergic signaling, renin-angiotensin-aldosterone signaling, and hypoxic signaling which may lead to dynamic miRNA expression changes in circulating inflammatory cells. Another possibility is that miRNA detected in the plasma or serum may be released directly from stressed organs and can influence miRNA expression in circulating cells which can then have systemic effects including in the myocardium in TOF again highlighting systemic transcriptional changes in patients with TOF even prior to the development of RV failure (Fig 6). We cannot assess the role of ACE inhibitors and angiotensin receptor blockers since none of the patients were on these agents. Although some of the patients were on β blockers, the cohort was too small to evaluate their role on the miRNA expression.

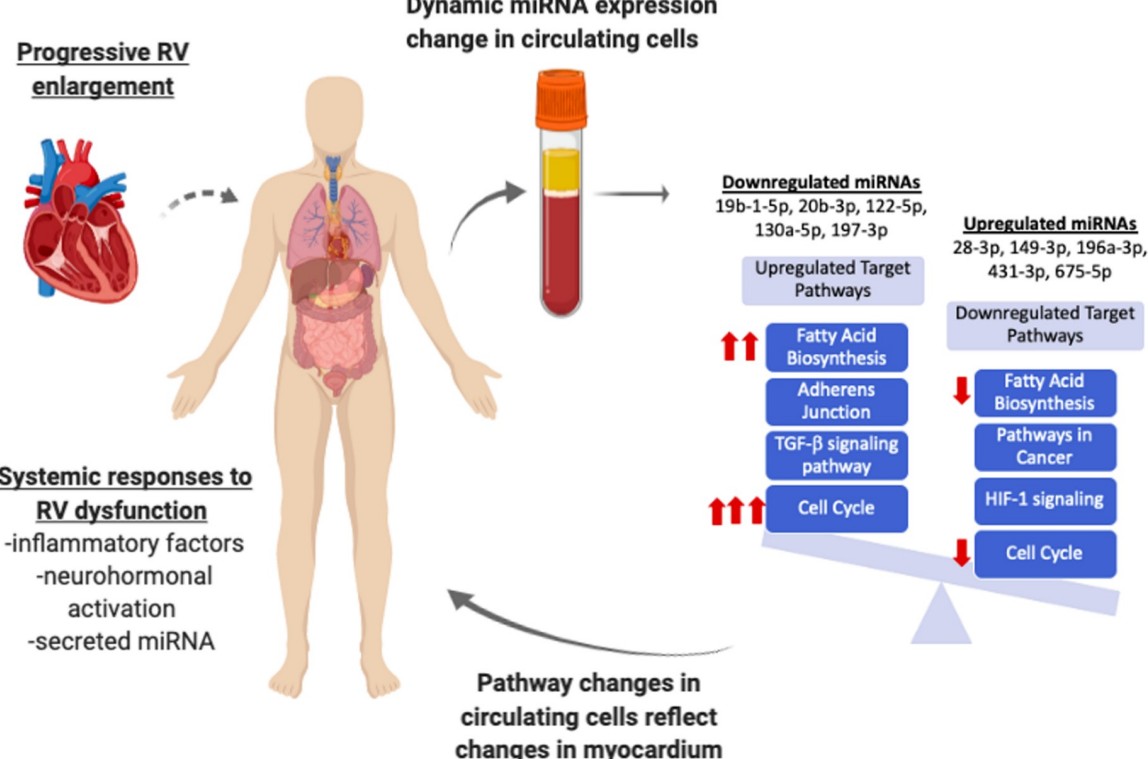

**Fig 6. Proposed schematic of progressive RV dysfunction leading to dynamic changes in circulating miRNA expression.** Progressive RV enlargement likely leads to systemic responses to incite global transcriptional changes within circulating cells which may subsequently incite changes in myocardial tissue, activating pathways including cell cycle, fatty acid biosynthesis, and TGF-β, while inhibiting HIF1. Red arrows denote the most dysregulated pathways.

We observed that global miRNA expression decreases with worsening RV enlargement. A similar finding has been seen in other reports evaluating miRNA signatures found in plasma of heart failure patients [17, 38] where the authors suggest that as patients develop worsening volume overload, a dilutional effect can lead to a reduction in measured plasma miRNA. Although this may play a role in miRNA measured in plasma, it seems less likely in the buffy coat. This reduction in total miRNA may lead to activation of a global mRNA transcriptional response to stress. Abu-Halima et al. [19] evaluated whole blood miRNA (circulating cells + plasma) in TOF patients with and without heart failure. They found a significant reduction in expression of miRNAs 421, 1233-3p, 625-5p in 3 patients with clinical heart failure. Although our findings did not validate their specifically reported miRNA, our work does corroborate that peripheral blood miRNA expression changes in adults with TOF and identifies a linear relationship between miRNA expression and progressive RV enlargement and dysfunction prior to the development of clinical heart failure. We also show that circulating miRNA expression can reclassify patients particularly those with mild-moderate RV enlargement into those with a miRNA signature similar to controls and those with miRNA signature similar to patients with severe RV enlargement. This data may be used to risk-stratify patients and determine timing of interventions. The increasing age across groups seen in our cohort may influence the peripheral blood miRNA. Since many of our patients did not have cardiac MRI data, it is not known whether MRI evaluation of RV size and function would better correlate with miRNA signature than echocardiographic evaluation of RV size and function.

We have previously shown dynamic miRNA changes in experimental RV failure with upregulation in cell death pathways, oxidant stress and extracellular matrix proteins and inhibition of angiogenic pathways in RV myocardium [18, 39]. Whether circulating miRNA signature provide insights into adverse RV remodeling or into the systemic response to RV stress is unknown. However, we now demonstrate circulating miRNA dysregulation which targets the very same pathways seen in our experimental animal model of RV failure suggesting that the circulating miRNA signature could indeed serve as a window into RV adverse remodeling (Fig 6).

Our findings provide insight into the potential role of multiple miRNA in mediating RV failure, including miRNAs 28, 371, 433, and 451. We have previously demonstrated that miRNA 28 is primarily increased in the non-myocyte fraction (predominantly fibroblasts) in RV failure [18]. The role of miRNA 28-3p in heart failure is unclear but interestingly miRNA 28-3p expression has been found to be positively correlated to acute coronary syndrome risk [40] and is increased in patients with diabetes [41]. MiRNA 28-3p is known to have an important role in regulating cell proliferation and differentiation [42–44] and may be used as a plasma biomarker of acute pulmonary embolism being released from damaged lung tissue [33]. MiRNAs 371, 433, and 451 have been reported to have important roles in cell proliferation, maintenance of redox state, and angiogenesis [30, 31, 45–50]. In summary these miRNAs regulate cell proliferation, enhance oxidative stress, and inhibit endothelial function, all of which can be used to follow disease progression and serve as future targets for RV specific drug development.

## Conclusion

Adults with repaired TOF are a rapidly growing group of patients with significant morbidity related to heart failure. Global miRNA expression in circulating inflammatory cells in the peripheral blood in repaired TOF patients identified upregulation in profibrotic pathways, cell proliferation and cell death pathways, and downregulation in fatty acid biosynthesis, and HIF-1α signaling. These observations were noted even prior to the development of clinical RV failure suggesting that many adverse remodeling pathways are active prior to the development of

changes seen by imaging. We describe for the first time unique mechanisms which may lead to RV failure and identify novel biomarkers which may be used to follow disease progression and aid in risk stratification.

## Supporting information

**S1 File.**
(PDF)

## Acknowledgments

We would like to thank Dr. George Lui of Stanford's Adult Congenital Heart Disease program for his thorough review and helpful comments for this manuscript.

## Author Contributions

**Conceptualization:** Chad S. Weldy, Saad Ali Syed, Rajesh Punn, Anne Taylor, Brittany Navarre, Sushma Reddy.

**Data curation:** Chad S. Weldy, Saad Ali Syed, Myriam Amsallem, Dong-Qing Hu, Xuhuai Ji, Sushma Reddy.

**Formal analysis:** Chad S. Weldy, Myriam Amsallem, Dong-Qing Hu, Xuhuai Ji, Rajesh Punn.

**Funding acquisition:** Sushma Reddy.

**Investigation:** Chad S. Weldy, Sushma Reddy.

**Methodology:** Chad S. Weldy, Saad Ali Syed, Anne Taylor, Brittany Navarre, Sushma Reddy.

**Project administration:** Chad S. Weldy, Sushma Reddy.

**Resources:** Sushma Reddy.

**Software:** Xuhuai Ji.

**Supervision:** Sushma Reddy.

**Validation:** Sushma Reddy.

**Writing – original draft:** Chad S. Weldy.

**Writing – review & editing:** Chad S. Weldy, Saad Ali Syed, Myriam Amsallem, Dong-Qing Hu, Xuhuai Ji, Rajesh Punn, Anne Taylor, Brittany Navarre, Sushma Reddy.

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
