## [Decision Letter · Decision Letter 0]

31 Jul 2020

PONE-D-20-18623

Circulating whole genome miRNA expression corresponds to progressive right ventricle enlargement and systolic dysfunction in adults with tetralogy of Fallot

PLOS ONE

Dear Dr. Weldy,

Thank you for submitting your manuscript to PLOS ONE. After careful consideration, we feel that it has merit but does not fully meet PLOS ONE’s publication criteria as it currently stands. Therefore, we invite you to submit a revised version of the manuscript that addresses the points raised during the review process.

Both reviewers believed that your manuscript was interesting and could be successfully revised with additional work. Both reviewers agreed that a major problem with the data presented in the manuscript was the interpretation of the results based on the controls chosen. Different controls as well as MRI data was suggested as approaches that would strengthen your manuscript. Please also address the other comments raised by the reviewers. If additional time is required to complete the revisions, the journal is likely to extend the deadline below.

We look forward to receiving your revised manuscript.

Kind regards,

Robert W Dettman, PhD

Academic Editor

PLOS ONE

Journal Requirements:

2.Thank you for stating the following in the Acknowledgments Section of your manuscript:

[SR was supported by grants from the NIH K08 HL127277, U.S. Department of Defense PR151448, Reddy Foundation, and American Heart Association 16GRNT31200008. MA was supported by a Stanford Cardiovascular Institute – MHRCI seed grant.]

 [The funders had no role in study design, data collection and analysis, decision to publish, or preparation of the manuscript.]

Reviewers' comments:

Reviewer's Responses to Questions

**Comments to the Author**

1. Is the manuscript technically sound, and do the data support the conclusions?

Reviewer #1: Partly

Reviewer #2: No

2. Has the statistical analysis been performed appropriately and rigorously? 

Reviewer #1: No

Reviewer #2: Yes

3. Have the authors made all data underlying the findings in their manuscript fully available?

Reviewer #1: Yes

Reviewer #2: No

4. Is the manuscript presented in an intelligible fashion and written in standard English?

Reviewer #1: Yes

Reviewer #2: Yes

5. Review Comments to the Author

Reviewer #1: I enjoyed reading the manuscript by Welby et al "Circulating whole genome miRNA expression corresponds to progressive right ventricle enlargement and systolic dysfunction in adults with tetralogy of Fallot"

The field is very interesting and potentially of significant clinical importance.

Nevertheless there are significant limitations to be addressed:

1) Tetralogy of Fallot is a large spectrum diagnosis varying from VSD with some aortic overriding and pulmonary/infundibular stenosis to complete pulmonary atresia. How can the authors we sure miRNA expression is also not affected by these diagnostic differences? For examples 3 patients had a late repair indicating a different anatomical and physiological RV substrate compared to the other 17 patients

2) It is not clear whether this was a prospective study with a specific protocol and inclusion/exclusion criteria or a retrospective study. How were the patients recruited? Was this a follow up study where patients were recruited in the out patient clinic?

3) A major limitation is the lack of MRI data which is possibly considered the goal standard for evaluating RV volumetrics and functions.

4) The sample timing was also very confusing. What do the authors mean for clinically indicated TEE? Were these routine follow up echos or indicated by specific symptoms related to a significant lesion (pulmonary valve regurgitation, stenosis, conduit dysfunction, RVOT arrhythmias, significant TR, etc). How could the authors exclude the possibility that changes in miRNA expression are not related to residual haemodynamic lesions?

5) Myocardial miRNA 28-3p was also measured in adult control RV (60 yrs of age) and adolescents (14 yrs) with HLHS. I am not sure these 2 groups are comparable and representative. They could have used RV biopsies taken at the time of surgery in adult TOF patients undergoing pulmonary valve replacement.

6) A sample size calculation is required as it is very difficult to draw any significant conclusion for comparing such small group of patients.

Reviewer #2: This original article entitled “Circulating whole genome miRNA expression corresponds to progressive right ventricle enlargement and systolic dysfunction in adults with tetralogy of Fallot” describes the circulating miRNA profile in adult patients who had TOF operation. The authors focused on three miRNAs: 28-3p, 433-3p and 371-3p. At least two miRNA’s expression showed a correlation to the degree of RV failure.

Major:

1. This reviewer was not convinced that miRNA profile is directly caused by RV failure. Although there was a correlation between the degrees of change in miRNA expression to RV failure, the degrees of change in miRNA may be related to the general changes in the patients’ health condition.

2. The authors concluded that miRNA-28-3p expression is increased in RV due to congenital heart disease in children comparing miRNA-28-3p expression between control adult hearts and child hearts with RV failure due to hypoplastic LV (Figure 6). To this reviewer, the current experimental design cannot reach this conclusion unless the authors use the control (adult) normal hearts. Although the authors discussed the limitation of this experiment (line 407), the result didn’t add any information. This section should be eliminated.

3. The discussion is too long, stating in broad terms the functions of three miRNAs, such as inflammation, cell cycle etc. The discussion did not compare the current study with the previous studies demonstrating circulating miRNA profile in congenital heart diseases (there were several citations). Did the authors find similar or different results? In addition, the discussion is too speculative, and the miRNA profile may not be specific to RV failure after TOF operation.

Minor:

The data deposit of the miRNA profile was not clearly mentioned.

6. PLOS authors have the option to publish the peer review history of their article (what does this mean?). If published, this will include your full peer review and any attached files.

Reviewer #1: No

Reviewer #2: No

---

## [Author Response · Author response to Decision Letter 0]

1 Sep 2020

Dear Dr. Dettman and invited reviewers,

Thank you for your consideration and careful review of our manuscript titled "Circulating whole genome miRNA expression corresponds to progressive right ventricle enlargement and systolic dysfunction in adults with tetralogy of Fallot". Here we submit a significantly modified manuscript, along with responses to each of the thoughtful reviewer comments. We hope our modified manuscript sufficiently addresses the important concerns highlighted from the reviewers.

Reviewer concerns and responses:

Reviewer 1:

1) Tetralogy of Fallot is a large spectrum diagnosis varying from VSD with some aortic overriding and pulmonary/infundibular stenosis to complete pulmonary atresia. How can the authors we sure miRNA expression is also not affected by these diagnostic differences? For examples 3 patients had a late repair indicating a different anatomical and physiological RV substrate compared to the other 17 patients

Response: We agree with the reviewer that tetralogy of Fallot is a complex congenital heart disease where not all patients are identical. We recognize that by the nature of performing research with congenital heart disease patients, there will always be limitations given the wide variation in patient anatomy. However, once these patients have undergone surgical palliation, the anatomical differences are reduced. Following surgical palliation, all patients are at risk for pulmonary insufficiency and stenosis and subsequent right ventricular (RV) dilation and dysfunction. Therefore, our study has some advantages in that it is now comparing only post-surgical patients allowing for similar comparisons between right ventricular (RV) function. Because of this, despite prior differences in anatomy and timing of repair, we are now able to more closely compare the difference in RV function as the primary differing factor between these patients.

Of the 3 patients with late repair, 1 patient was in the “normal RV size” group, and 2 patients were in the “mild/moderate RV enlargement” group. That these patients did not lie entirely at one end of the spectrum of RV size and function (i.e. all in the “normal size” group or all in the “severe enlargement” group) is supportive that they are not unique outliers in their response to RV stress. This data on these 3 patients has now been added to the manuscript [Page 10, Line 236-238].

2) It is not clear whether this was a prospective study with a specific protocol and inclusion/exclusion criteria or a retrospective study. How were the patients recruited? Was this a follow up study where patients were recruited in the out patient clinic?

Response: In this study, we hypothesized that miRNA would be dysregulated in patients with TOF with varying RV size and function.

We defined a patient population as noted in inclusion criteria, and then recruited patients that fit that the inclusion criteria in a prospective manner in an outpatient setting. All patients with TOF are routinely followed on a yearly basis by their cardiologist since many are prone to pulmonary insufficiency and pulmonary stenosis and are at risk for RV failure. All blood samples were also collected during their outpatient clinic visit once they consented to the study. We have updated the methods section to clearly note our methods of sample collection [Page 6, Line 103 – 107].

3) A major limitation is the lack of MRI data which is possibly considered the goal standard for evaluating RV volumetrics and functions.

Response: We agree that cardiac MRI (cMRI) is the gold standard for evaluating RV size and function. There is currently no clearly defined criteria as to when and how frequently a patient with TOF following repair should undergo cMRI. While all patients who are being considered for pulmonary valve replacement undergo a cMRI, those with no RV dilation or mild to moderate dilation do not routinely undergo cMRI or undergo cMRI every 5 years. Therefore, many of our patients did not have a cMRI at the time of recruitment and blood samples collection. However, all patients had an echocardiogram at the time of recruitment and blood draw and therefore RV size and function by echocardiogram was used to correlate with circulating microRNA expression. 

Of the 20 patients included in our study, 12 patients had a cMRI but there were only 4 patients where cMRI was done at or near the time of echocardiogram and blood draw and are included in the table below. The RVEDA (cm2), RVESA (cm2) and RVFAC (%) by echocardiogram were compared to the MRI RVEDV (ml), RVESV (ml) and RVEF (%) respectively. There is a strong and significant correlation between the echocardiogram and the cMRI assessment of RV size (measures of RVEDA or RVESA vs RVEDV or RVESV respectively: R=0.83, p<0.01) and a trend of association for measure of RV systolic function (RVFAC vs RVEF: R=0.77, p=0.22). We recognize that the lack of cMRI is a limitation but we show here that the echocardiograms also provide excellent insight into RV size and function at the time of blood draw and miRNA profile analysis. We have modified the discussion to further highlight the limitations in the lack of cMRI. 

 TTE Cardiac MRI 

TTE Group RVEDA (cm2) RVESA (cm2) RVFAC RVEDV (ml) RVESV (ml) RVEF %

Mild/Mod 43 27 37 211 130 39

Mild/Mod 36 17 52 237 145 39

Mild/Mod 33 20 41 240 146 39

Severe 52 37 28 285 180 37

A paragraph in the methods section and results section has been added which addresses this concern:

Page 7, line 134-140. Cardiac MRI (cMRI) imaging was not available for all patients and therefore comparisons of RV size and function to miRNA signature based on cMRI were not performed. Of 20 patients recruited, 12 patients had a cMRI but there were only 4 patients where cMRI was done at or near the time of echocardiogram and blood draw (Supplemental Table 1). Therefore, assessment of RV size and function based on TTE imaging done at the time of blood sample collection was used. RV size and function by TTE was compared with the gold standard cMRI derived RV size and function in the 4 patients with recently available cMRI. 

Page 10, line 243-248. The RVEDA (cm2), RVESA (cm2) and RVFAC (%) by echocardiogram were compared to the MRI RVEDV (ml), RVESV (ml) and RVEF (%) respectively in 4 patients where cMRI was available. There is a strong and significant correlation between the echocardiogram and the cMRI assessment of RV size (measures of RVEDA or RVESA vs RVEDV or RVESV respectively: R=0.83, p<0.01) and a trend of association for measure of RV systolic function (RVFAC vs RVEF: R=0.77, p=0.22) (Supplemental Table 1).

4) The sample timing was also very confusing. What do the authors mean for clinically indicated TEE? Were these routine follow up echos or indicated by specific symptoms related to a significant lesion (pulmonary valve regurgitation, stenosis, conduit dysfunction, RVOT arrhythmias, significant TR, etc). How could the authors exclude the possibility that changes in miRNA expression are not related to residual haemodynamic lesions?

Response: All patients with TOF are routinely followed on a yearly basis by their cardiologist since many are prone to pulmonary insufficiency and pulmonary stenosis and are at risk for RV failure. This yearly visit (which we are calling clinically indicated and not a research related visit) includes a symptom assessment, physical examination, electrocardiogram and transthoracic echocardiogram (TTE). This is the current national standard of care recommended by the American College of Cardiology. Patients were recruited during these visits if they met the inclusion criteria and blood samples were collected on the same day and correlated with the TTE performed also on that day. The TTEs were not performed for any specific symptoms but as part of routine care. We have now clarified this in the ‘Methods’ section.

We agree with the reviewer that the hemodynamic load leading to RV stress, in patients with no other organ system dysfunction is what is reflected in the circulating microRNA signature. Indeed, patients can have a similar degree of pulmonary insufficiency and RV stress by echocardiogram but have a different circulating microRNA signature which can further differentiate patients into those who many need earlier intervention. This would be an excellent clinical adjunct to TTEs and cMRIs for the cardiologist. We have now added this to the ‘Discussion’ section.

5) Myocardial miRNA 28-3p was also measured in adult control RV (60 yrs of age) and adolescents (14 yrs) with HLHS. I am not sure these 2 groups are comparable and representative. They could have used RV biopsies taken at the time of surgery in adult TOF patients undergoing pulmonary valve replacement.

Response: We agree that there are significant limitations to this comparison. In TOF, the majority of adults undergoing PVR do so for pulmonary insufficiency, where no muscle is routinely collected at the time of surgery. Samples such as that would be ideal, but unfortunately, we do not have access to such samples. We find our presented data quite intriguing given there is consistency between our peripheral blood data, prior mouse data evaluating RV failure, and our current evaluation of the RV from patients with HLHS. But, we agree that the limitations of this comparison are likely too great to overcome and may provide confusion to a reader. We will remove figure 6.

6) A sample size calculation is required as it is very difficult to draw any significant conclusion for comparing such small group of patients.

The reviewer brings up an important question regarding the size of the study performed. We address this limitation in our manuscript and highlight that this was an exploratory study. We do agree that multiple comparison testing with microRNA array data is crucial. Importantly, we address multiple comparison testing concerns with miRNA arrays by performing FDR calculation for analysis.

Reviewer 2:

1) This reviewer was not convinced that miRNA profile is directly caused by RV failure. Although there was a correlation between the degrees of change in miRNA expression to RV failure, the degrees of change in miRNA may be related to the general changes in the patients’ health condition.

Response: We agree with the reviewer that the degree of change in miRNA may be related to the general changes in the patients’ health condition and or RV failure. In order to address this important point, we recruited patients with similar cardiac and extracardiac organ system status. First, cardiac health status was similar between each of these 20 patients as assessed by NYHA classification (all patients with NYHA class I or II symptoms, no patient was in NYHA III or IV). Importantly, we wanted to exclude further factors such as renal failure or hepatic injury secondary to progressive RV failure. We reported in Table 1 the lab values of sodium, Cr, AST/ALT/Alk Phos, T bili, each of which were within normal limits and not different between groups. We also evaluated BMI, which was found to be similar between groups. Given this, these 20 patients appear from a clinical and laboratory standpoint to be quite similar. We acknowledge that other factors which may conceivably modify peripheral blood miRNA expression, such as exercise and diet, were not taken into account. This has now been included as a limitation to our study.

2) The authors concluded that miRNA-28-3p expression is increased in RV due to congenital heart disease in children comparing miRNA-28-3p expression between control adult hearts and child hearts with RV failure due to hypoplastic LV (Figure 6). To this reviewer, the current experimental design cannot reach this conclusion unless the authors use the control (adult) normal hearts. Although the authors discussed the limitation of this experiment (line 407), the result didn’t add any information. This section should be eliminated.

Response: We agree with the major limitations to presenting this data. As discussed above in response to a similar concern brought by reviewer 1, we will remove this data.

3) The discussion is too long, stating in broad terms the functions of three miRNAs, such as inflammation, cell cycle etc. The discussion did not compare the current study with the previous studies demonstrating circulating miRNA profile in congenital heart diseases (there were several citations). Did the authors find similar or different results? In addition, the discussion is too speculative, and the miRNA profile may not be specific to RV failure after TOF operation.

Response: We thank the reviewer for this comment. The discussion has been abbreviated and further emphasis on the comparison to the prior reported studies has been added. In general, the prior studies did not evaluate miRNA expression of the circulating cells, and the one prior study which included comparison of RV function had only 3 patients with heart failure, and had evaluated miRNA in whole blood. Therefore, it is not surprising that dysregulated miRNAs in their study were not replicated in ours.

We agree that it is difficult to fully know if a change in miRNA profile is specific to changes in the RV versus an alternative unmeasured factor. As noted above in response to review two’s question 1, we attempted to evaluate for other factors and highlight that in these 20 patients, they are largely similar in terms of clinical status and laboratory data. The miRNA profiles seem to be associated with changes in RV size and function. Future studies will focus on evaluating miRNA profile changes over time within individual patients which may help to answer these questions.

Updated text in discussion:

Page 17-18, Line 406-420 now reads as follows: We observed that global miRNA expression decreases with worsening RV enlargement. A similar finding has been seen in other reports evaluating miRNA signatures found in plasma of heart failure patients (1, 2) where the authors suggest that as patients develop worsening volume overload, a dilutional effect can lead to a reduction in measured plasma miRNA. Although this may play a role in miRNA measured in plasma, it seems less likely in the buffy coat. This reduction in total miRNA may lead to activation of a global mRNA transcriptional response to stress. Abu-Halima et al. (3) evaluated whole blood miRNA (circulating cells + plasma) in TOF patients with and without heart failure. They found a significant reduction in expression of miRNAs 421, 1233-3p, 625-5p in those with clinical heart failure, but was very limited in patient numbers and evaluation of RV size and function (TOF without HF: N = 34, TOF with HF: N = 3). Although our findings did not validate their specifically reported miRNA, our work does corroborate that peripheral blood miRNA expression changes in adults with TOF and identifies a linear relationship between miRNA expression and progressive RV enlargement and dysfunction prior to the development of clinical heart failure.

Minor:

The data deposit of the miRNA profile was not clearly mentioned.

Response: Data is in process of being uploaded onto GEO to be made publicly available.

REFERENCES

1. Watson CJ, Gupta SK, O'Connell E, Thum S, Glezeva N, Fendrich J, et al. MicroRNA signatures differentiate preserved from reduced ejection fraction heart failure. Eur J Heart Fail. 2015;17(4):405-15.

2. Bilchick K, Kothari H, Narayan A, Garmey J, Omar A, Capaldo B, et al. Cardiac resynchronization therapy reduces expression of inflammation-promoting genes related to interleukin-1beta in heart failure. Cardiovasc Res. 2019.

3. Abu-Halima M, Meese E, Keller A, Abdul-Khaliq H, Radle-Hurst T. Analysis of circulating microRNAs in patients with repaired Tetralogy of Fallot with and without heart failure. J Transl Med. 2017;15(1):156.

---

## [Decision Letter · Decision Letter 1]

16 Oct 2020

Circulating whole genome miRNA expression corresponds to progressive right ventricle enlargement and systolic dysfunction in adults with tetralogy of Fallot

PONE-D-20-18623R1

Dear Dr. Weldy,

We’re pleased to inform you that your manuscript has been judged scientifically suitable for publication and will be formally accepted for publication once it meets all outstanding technical requirements. Sorry for the delays, one of the reviewers had some issues.

Kind regards,

Robert W Dettman, PhD

Academic Editor

PLOS ONE

Additional Editor Comments (optional):

Reviewers' comments:

Reviewer's Responses to Questions

**Comments to the Author**

1. If the authors have adequately addressed your comments raised in a previous round of review and you feel that this manuscript is now acceptable for publication, you may indicate that here to bypass the “Comments to the Author” section, enter your conflict of interest statement in the “Confidential to Editor” section, and submit your "Accept" recommendation.

Reviewer #1: All comments have been addressed

2. Is the manuscript technically sound, and do the data support the conclusions?

Reviewer #1: Yes

3. Has the statistical analysis been performed appropriately and rigorously? 

Reviewer #1: Yes

4. Have the authors made all data underlying the findings in their manuscript fully available?

Reviewer #1: Yes

5. Is the manuscript presented in an intelligible fashion and written in standard English?

Reviewer #1: Yes

6. Review Comments to the Author

Reviewer #1: The authors have better highlighted the limitation of this study. Considering the population we are dealing with, I feel this study is clinically relevant and could potentially lead to a much bigger study.

7. PLOS authors have the option to publish the peer review history of their article (what does this mean?). If published, this will include your full peer review and any attached files.

Reviewer #1: No

---

## [Editor Report · Acceptance letter]

20 Oct 2020

PONE-D-20-18623R1 

Circulating whole genome miRNA expression corresponds to progressive right ventricle enlargement and systolic dysfunction in adults with tetralogy of Fallot 

Dear Dr. Weldy:

I'm pleased to inform you that your manuscript has been deemed suitable for publication in PLOS ONE. Congratulations! Your manuscript is now with our production department. 

Kind regards, 

on behalf of

Dr Robert W Dettman 

Academic Editor

PLOS ONE